# Geometric Inductive Priors in Diffusion-Based Optical Flow Estimation

Alberto Pepe, Joan Lasenby
Probabilistic Systems,
Information, and Inference Group
University of Cambridge
Trumpington Street, Cambridge, CB21PZ, UK

`{ap2219, jl221}@cam.ac.uk`

Paulo dos Santos Mendonca
Applied Sciences Group
Microsoft
One Microsoft Way, Redmond, WA 98052

`paulo.dos@microsoft.com`

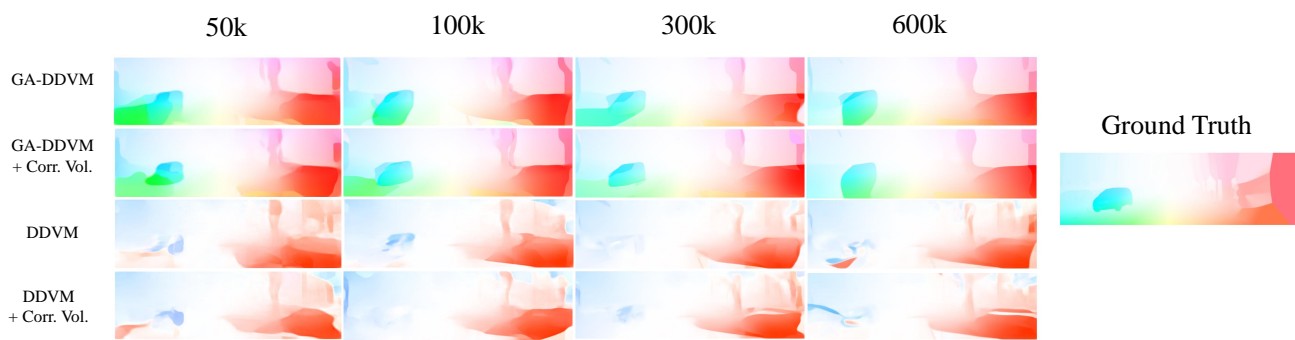

Figure 1. **Constraining diffusion models to operate within the representational space of the target quantity can substantially improve convergence speed**. Ground truth and estimated optical flow for frame 0 in the KITTI dataset, visualized across four different architectures trained for an increasing number of steps. **GA-DDVM** (ours) produces realistic optical flow as early as 50,000 steps into training by constraining the diffusion pipeline to learn only rotations and scalings of 2D vector fields. Code can be found here.

## Abstract

*Diffusion models are ubiquitous in generative modeling and their prevalence in structured prediction tasks is increasing. The denoising diffusion vision model (DDVM), for example, achieves state-of-the-art accuracy on tasks such as monocular depth and optical flow estimation. We introduce GA-DDVM, a modified version of DDVM working in Geometric Algebra (GA) that includes a geometric prior to constrain diffusion for faster and more accurate optical flow estimation. We constrain diffusion in two key ways: (i) we restrict the types of objects learned by the pipeline to 2D vector fields, (i.e., optical flows), and (ii) we limit the operations performed by the network layers on these objects to scaling and rotations. GA-DDVM demonstrates substantial improvements over the baseline DDVM that emerge early in training and persist across all checkpoints: at 600k training steps, GA-DDVM reduces the endpoint error (EPE) on the KITTI dataset by 76.3% and reduces the KITTI Fl-all metric from 76.8% to 20.1%. The Sintel-clean error and Sintel-final errors similarly drop from 11.4 to 3.38, and from 11.7 to 4.46, respectively. By embedding geometric struc-*
*ture directly into the diffusion process, GA-DDVM shows that incorporating domain priors into generative models can yield substantially faster convergence with minimal additional complexity in network architecture. This opens up promising directions for structured prediction tasks across domains where geometric constraints are inherent.*

## 1. Introduction

Diffusion models have become the leading approach for a wide range of generative tasks, including image and video synthesis [19, 20], 3D molecular structure generation [66, 67], text-to-image translation [56, 57], and virtual environment generation [37, 52].

Despite their versatility, diffusion models have several drawbacks: they are computationally expensive, require larger and more diverse datasets compared with alternatives, converge slowly, and lack interpretability due to their multi-step stochastic nature [15, 65].

We address these limitations with a unified and conceptually simple strategy. We focus on optical flow estimation,

building on the DDVM pipeline from [58], and introduce **GA-DDVM**. The premise is straightforward: if the goal is to estimate 2D vector fields, such as optical flow, then the diffusion process should reason in terms of 2D vectors. In other words, models should be constrained in terms of *what* they learns, i.e., 2D vector fields, and *how* they learn, i.e., through operations that are geometrically meaningful in this domain, such as scaling and rotations in 2D. Our contributions are as follows:

- We introduce the first diffusion pipeline that operates within the framework of Geometric Algebra, providing a simple yet powerful geometric prior for optical flow estimation.
- By constraining the training process to focus exclusively on 2D vector fields, we achieve significantly faster convergence, reducing the training end-point error (EPE) from 17.0 with DDVM to 3.99 with GA-DDVM in just 50,000 steps.
- As a result, we significantly reduce training time and extract more information from fewer data, enabling accurate diffusion even with limited computational resources or data.
- By working explicitly in 2D space, inputs, outputs, activations, weights and biases can all be visualized as objects or operations in 2D space, enabling interpretability.

Embedding both data and architecture within a consistent framework provided by Geometric Algebra offers a simple yet powerful strategy for designing diffusion pipelines that are fast-converging, interpretable, and grounded in the geometry of the problem.

## 2. Related Work

**Optical flow estimation**. Optical flow estimation has progressed from classical variational and local methods to modern deep learning approaches. Early algorithms such as Horn–Schunck [21] and Lucas–Kanade [33] framed the problem as an optimization task based on assumptions like brightness constancy and spatial smoothness. To address larger displacements, multi-scale strategies [2, 3] and coarse-to-fine warping schemes [4] were developed. Feature-based approaches, including SIFT Flow [32] and DeepFlow [64], improved robustness to appearance changes by leveraging dense descriptors and explicit matching costs.

The advent of deep learning began with FlowNet [14], which demonstrated that convolutional networks can learn to predict dense flow directly from image pairs. Subsequent models such as SpyNet [53], PWC-Net [61], LiteFlowNet [22], and IRR-PWC [23] integrated classical principles—pyramidal representations, warping, and cost volumes—into efficient neural architectures. RAFT [63] further improved accuracy through dense correlation volumes

and recurrent refinement, achieving state-of-the-art results. More recent developments explore unsupervised [25, 36], self-supervised [24], and transformer-based frameworks [68, 70], reflecting a shift toward combining geometric reasoning with learned representations.

Throughout this evolution, the integration of domain knowledge, such as smoothness priors, multi-scale structure, and geometric constraints, has remained central to effective optical flow estimation.

**Diffusion models**. Diffusion models are a class of generative methods, initially introduced through denoising score matching and stochastic differential equations [19, 59, 60]. These models generate data by reversing a gradual process where noise is added to inputs until they become indistinguishable from pure noise. In computer vision, diffusion-based architectures have demonstrated remarkable success in tasks such as image generation, editing, and synthesis [9, 10, 38, 51, 69].

The Denoising Diffusion Vision Model (DDVM) [58] extends diffusion models to dense prediction tasks such as monocular depth and optical flow estimation. Unlike conventional regression-based approaches, DDVM frames these tasks probabilistically, modeling dense outputs as samples from a learned distribution conditioned on input frames. This probabilistic formulation enables the model to resolve inherent ambiguities and produce robust estimates even in challenging scenarios such as in the presence of occlusions, where classical heuristics often fail. A second key distinction lies in DDVM's simplicity: it eliminates the need for problem-specific choices in the model architecture and hand-crafted heuristics that have historically accumulated in dense prediction pipelines. Instead, it offers a unified, clean, task-agnostic architecture applicable across domains.

Although DDVM marks a significant and scalable advance toward general-purpose structured prediction in vision, its computational demands remain high, making it impractical for real-time optical flow estimation.

**Geometric Algebra (GA) networks**. GA networks are a class of hypercomplex neural networks that apply the mathematical framework of Clifford (or Geometric) Algebra to represent and process geometric relationships more effectively [5, 6, 39, 40, 55, 72]. GA extends linear algebra by providing a unified system for representing scalars, vectors, complex numbers, quaternions, and higher-order geometric entities. It enables concise formulations of transformations such as rotations, reflections, and incidence, which are otherwise cumbersome in standard vector spaces [12, 13, 18, 27, 28, 30].

These properties make GA a versatile tool across domains such as molecular modeling [1, 31, 43–45], camera

pose estimation [42, 47], rotation estimation [29, 34, 48], 3D alignment [35, 50], and modeling of partial differential equations (PDEs) [5, 46, 49, 71]. In [17], CliffPhys is introduced for the joint estimation of optical flow and depth estimation.

GA networks are *model-agnostic* and can be integrated into a wide range of architectures, including ResNets [5, 46], graph neural networks [72], neural operators [5, 49], and transformers [6]. These GA-enhanced models preserve the original architectural advantages while enhancing them with expressive, geometry-aware computations. For example, GA layers can be made steerable and equivariant [47, 54, 55, 71], using geometric and sandwich product layers. Such layers act as *learnable geometric templates*, enabling networks to capture task-relevant spatial patterns within the algebraic framework.

In practice, building a network in GA requires two simple steps:

- **Choose the appropriate algebra.** Select a geometric algebra $G(p, q, r)$ suited to the task, defined by its signature and dimension $n = p + q + r$. This determines the multivector structure, available products, and semantics of computation. Examples include $G(n, 0, 0)$ for $n$D Euclidean space, $G(1, n, 0)$ for $n$D spacetime, $G(n, 1, 0)$ for projective geometry, and $G(n+1, 1, 0)$ for $n$D conformal geometry.
- **Embed data into the algebra.** Map inputs into the multivector space in a way that aligns with the structure of the algebra. The embedding defines how the network interprets and manipulates geometric information.

These two steps define the inductive bias and geometric reasoning capacity of the model. A well-chosen algebra and embedding are key to leveraging the full potential of GA in learning systems.

## 3. Method

### 3.1. Definion of Optical Flow

Images are maps from $X = \{0, \ldots, W - 1\} \times \{0, \ldots, H - 1\}$, where $H$ and $W$ are the images width and height, to $C = \{0, \ldots, 2^{b-1}\}^d$, where $b$ is number of bits, and $d = 1$ for grayscale, $d = 3$ for RGB, $d = 4$ for RGBA (RBG plus an alpha channel), etc. $X$ is geometric lattice of integer pixel coordinates, and $C$ is the photometric space of integer pixel intensities. In this way, an image is a map $I$:

$$I : X \to C \tag{1}$$
$$\mathbf{x} \mapsto c = I(\mathbf{x}) \tag{2}$$

with domain $X \subset \mathbb{R}^2$ (a rectangle) and co-domain $\mathbb{R}^d$. We refer to the value of $c$ of $I$ at $\mathbf{x}$ by the "color" of the pixel $\mathbf{x} = (x, y)$. The set of such maps $I$ is denoted by $\mathcal{I}$.

Let $I_0, I_1 : X \to C$ be two images observed at consecutive time steps $\tau, \tau + \delta\tau$, respectively. The *optical flow* is a function

$$\mathbf{v} : X \to \mathbb{R}^2 \tag{3}$$

which assigns a displacement vector $\mathbf{v}(\mathbf{x})$ to each pixel $\mathbf{x} \in X$, representing the apparent motion of that pixel from image $I_0$ to image $I_1$. That is,

$$\mathbf{v}(\mathbf{x}) = \begin{pmatrix} u(\mathbf{x}) \\ v(\mathbf{x}) \end{pmatrix} \tag{4}$$

This vector field defines how a point $\mathbf{x} \in X$ in image $I_0$ moves to its new location in image $I_1$. The *brightness constancy assumption* is expressed as

$$I_0(\mathbf{x}) = I_1(\mathbf{x} + \mathbf{v}(\mathbf{x})), \tag{5}$$

It relies on several assumptions:

1. **Noise-free imaging:** Images are acquired without noise. In reality, sensor noise and quantum effects violate this condition.
2. **Perfect flow:** The flow vector $\mathbf{v}(\mathbf{x})$ is exact. In practice, it is estimated from noisy data.
3. **Lambertian surfaces:** Surfaces reflect light uniformly in all directions. This fails for specular, transparent, or glossy materials.
4. **No occlusions or disocclusions:** The scene is equally visible in both frames. Occlusions and disocclusions (due to motion or viewpoint change, for exmaple,) lead to violations of this assumption.

The probabilistic approach offered by diffusion models such as DDVM and GA-DDVM is able to account for the uncertainty in the flow vectors by modeling them as random variables. This allows the model to accommodate noise, occlusions, and other imperfections in real-world data, providing a more robust solution compared to deterministic methods that rely on more strict assumptions.

### 3.2. Diffusion models

Diffusion models, possibly conditioned on an external signal $\xi$, aim to generate samples from a target distribution over a random variable $\mathbf{y} \in \mathbb{R}^d$, denoted $\mathbf{y} \sim p_\infty(\mathbf{y})$. To achieve this, the process begins with a simple prior distribution $\mathbf{y}_0 \sim p_0(\mathbf{y})$, typically a standard multivariate Gaussian, from which samples are easily drawn. A parametric transformation $F : \Theta \times \mathbb{R}^d \to \mathbb{R}^d$ is then applied iteratively to evolve the distribution of $\mathbf{y}_0$ toward the target. That is, given a sequence of parameters $\theta_1, \ldots, \theta_T$, the process defines a sequence of random variables:

$$\mathbf{y}_t = F(\theta_t, \mathbf{y}_{t-1}), \quad t = 1, \ldots, T, \tag{6}$$

such that the distribution of $\mathbf{y}_T$ approximates that of $\mathbf{y}_\infty$. The sequence $\{\mathbf{y}_t\}_{t=0}^T$ forms a stochastic process, i.e. an indexed collection of random variables over $\mathbb{R}^d$.

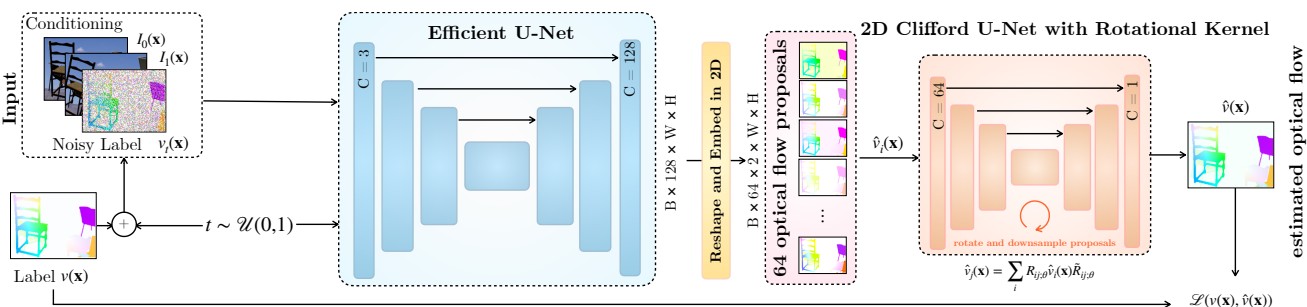

Figure 2. Architecture of GA-DDVM, which employs a W-Net backbone composed of two cascaded U-Nets. The first is the Efficient U-Net from the original DDVM pipeline (with its final convolutional layer removed), comprising 700M parameters. The second is a lightweight, specialised U-Net with 60M parameters, built from steerable geometric layers [55] operating in the geometric algebra $G(2,0,0)$. Its convolutional layers perform only learnable scalings and rotations on the input flow vectors $\hat{\mathbf{v}}_i(\mathbf{x})$ using sandwich product layers, promoting geometric consistency and accelerating convergence.

To train diffusion models, a denoising objective is typically used. The training process assumes access to samples $(\xi, \mathbf{y})$ from a joint data distribution, and uses a noise schedule $\gamma_t \in [0, 1]$ to define intermediate noisy versions of $\mathbf{y}$. Specifically, a noisy sample $\mathbf{y}_t$ is constructed as:

$$\mathbf{y}_t = \sqrt{\gamma_t}\,\mathbf{y} + \sqrt{1 - \gamma_t}\,\boldsymbol{\epsilon}, \quad \boldsymbol{\epsilon} \sim \mathcal{N}(0, I) \qquad (7)$$

The model $f_\theta$ is trained to predict the added noise $\boldsymbol{\epsilon}$ from $\mathbf{y}_t$, the conditioning variable $\xi$, and the timestep $t$. The loss function minimized during training is:

$$\mathbb{E}_{(\mathbf{x},\mathbf{y})}\,\mathbb{E}_{(t,\boldsymbol{\epsilon})} \left\| f_\theta\left(\mathbf{x}, \underbrace{\sqrt{\gamma_t}\,\mathbf{y} + \sqrt{1 - \gamma_t}\,\boldsymbol{\epsilon}}_{\mathbf{y}_t}, t\right) - \boldsymbol{\epsilon} \right\|_2^2 \qquad (8)$$

This denoising score matching objective encourages the network to recover the noise component, effectively learning to reverse the diffusion process.

In our case, the target variable $\mathbf{y}$ is the ground-truth optical flow $\mathbf{v}(\mathbf{x})$, and the conditioning variable $\xi$ is the pair of input images $(I_0, I_1)$. That is, the diffusion model is trained to sample optical flow fields conditioned on the observed image pair, as done in [58]. During training, the network learns to reconstruct the noise added to $\mathbf{v}(\mathbf{x})$ from $\mathbf{v}_t(\mathbf{x})$ thereby learning a generative model for optical flow consistent with the input frames.

### 3.3. The $G(2,0,0)$ Space

The geometric algebra $G(2,0,0)$, which we employ for the task of optical flow estimation, represents the Clifford algebra over the 2-dimensional Euclidean space $\mathbb{R}^2$ with a positive-definite metric. This algebra is generated by an orthonormal basis $\{e_1, e_2\}$ that satisfies the relations:

$$e_1^2 = e_2^2 = 1, \quad e_1 e_2 = -e_2 e_1.$$

The full algebra consists of multivectors, which are formed by the linear span of the basis elements:

$$G(2,0,0) = \mathrm{span}_{\mathbb{R}}\{1, e_1, e_2, e_{12}\}, \quad \text{where } e_{12} := e_1 e_2.$$

The element $e_{12}$ is a unit bivector and satisfies:

$$e_{12}^2 = -1,$$

indicating that it represents the oriented area element in $\mathbb{R}^2$, analogous to the imaginary unit $i$ in the complex plane. This structure allows for concise and geometrically intuitive representations of transformations such as rotations.

In the context of optical flow, we model flow vectors as elements in $\mathbb{R}^2 \subset G(2,0,0)$. A flow vector $\mathbf{v}(\mathbf{x}) \in \mathbb{R}^2$ is represented as:

$$\mathbf{v}(\mathbf{x}) = v(\mathbf{x})e_1 + u(\mathbf{x})e_2,$$

where $v$ and $u$ are the flow components in the $e_1$ and $e_2$ directions, respectively. A rotation of the flow vector by an angle $\theta$ is performed using the *sandwich product*, which is defined as:

$$\mathbf{v}' = R v R^{-1},$$

where $R(\theta) = \cos\left(\frac{\theta}{2}\right) - \sin\left(\frac{\theta}{2}\right) e_{12}$ is the rotor representing the rotation, and $R^{-1} = \cos\left(\frac{\theta}{2}\right) + \sin\left(\frac{\theta}{2}\right) e_{12}$ is its inverse, which is equal to the reverse of $R$, denoted $\widetilde{R}$.

## 4. Architecture: GA-DDVM

Our proposed architecture, GA-DVVM, is shown in Fig. 2. GA-DDVM relies on a W-Net backbone, i.e. two cascaded U-Nets: the first U-Net is the Efficient U-Net of the DDVM pipeline without its last convolutional layer, consisting of about 700 million trainable parameters. The second U-Net is a much smaller, yet specialised network, consisting of 60 million parameters.

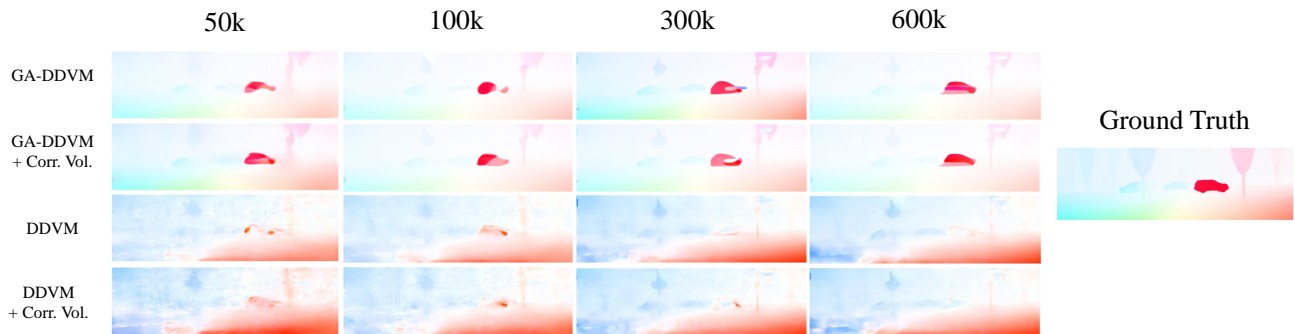

Figure 3. Ground truth and estimated optical flow for frame 10 in the KITTI dataset through four different architectures trained for an increasing number of steps.

Table 1. Error with (a) DDVM (b) DDVM + Corr. Vol. (c) GA-DDVM (ours) (d) GA-DDVM + Corr. Vol. (ours) vs training steps. The four pipeliens have been trained from scratch on AutoFlow and evaluated zero-shot on KITTI and Sintel at the reported checkpoints.

| | 50k | | | | | 100k | | | | | 300k | | | | | 600k | | | | |
|---|---|---|---|---|---|---|---|---|---|---|---|---|---|---|---|---|---|---|---|---|
| | AF | K. EPE | K. Fl-all | S. cl. | S. f. | AF | K. EPE | K. Fl-all | S. cl. | S. f. | AF | K. EPE | K. Fl-all | S. cl. | S. f. | AF | K. EPE | K. Fl-all | S. cl. | S. f. |
| (a) | 17.0 | 27.4 | 76.8 | 11.4 | 11.7 | 18.6 | 26.6 | 76.8 | 11.3 | 11.9 | 14.8 | 25.8 | 29.0 | 10.9 | 10.9 | 13.3 | 24.9 | 27.6 | 11.1 | 11.4 |
| (b) | 18.0 | 28.1 | 77.8 | 11.7 | 12.0 | 18.6 | 27.0 | 76.7 | 11.6 | 11.9 | 15.1 | 25.7 | 29.4 | 10.9 | 11.3 | 12.4 | 25.3 | 28.6 | 10.9 | 11.3 |
| (c) | 3.39 | **10.1** | 34.4 | **4.31** | **5.33** | **2.20** | **8.72** | **29.0** | 3.96 | 4.99 | **1.06** | 6.62 | 22.3 | 3.38 | 4.46 | 0.87 | 5.89 | 20.1 | **3.07** | **3.82** |
| (d) | **2.66** | 10.3 | **33.8** | 4.53 | 5.53 | 2.59 | 8.94 | 29.4 | **3.79** | **4.83** | 1.66 | **6.34** | **21.8** | **3.21** | **4.22** | **0.79** | **5.80** | **19.6** | 3.15 | 3.91 |

The second U-Net is built from steerable layers of [55], it works in $G(2,0,0)$ and its convolutional layers can only perform scaling and rotations, i.e. they implement a learnable transformation $\Phi_\theta(\cdot)$ over input flows $\hat{\mathbf{v}}_i(\mathbf{x})$ of the type:

$$\hat{\mathbf{v}}_j(\mathbf{x}) = \Phi_\theta(\hat{\mathbf{v}}_i(\mathbf{x})) = \sum_{i=1}^{C} \alpha_{ij;\theta} R_{ij;\theta} \hat{\mathbf{v}}_i(\mathbf{x}) \tilde{R}_{ij;\theta} \qquad (9)$$

with $\alpha_i \in \mathbb{R}$, $C$ the number of channels and $R_i$ rotors in $G(2,0,0)$. We also add a vector-valued bias term, $b_{ij;\theta}$, to Eq. 9.

The input to the second U-Net is the reshaped output of the first U-Net: instead of employing a simple convolutional layer to map 128 channels down to 2 vector channels of the optical flow, as in DDVM, we process them as follows:

- We remove the last convolutional layer from the Efficient U-Net.
- We reshape the tensor that would have been fed into the last convolutional layer, $x \in \mathbb{R}^{B \times 128 \times W \times H}$, into $x \in \mathbb{R}^{B \times 64 \times 2 \times W \times H}$. We refer to these as optical flow "proposals" $\hat{\mathbf{v}}(\mathbf{x})$.
- We embed the third dimension of size 2 as the vector components of a geometric algebra tensor with shape $\mathbb{R}^{B \times 64 \times 4 \times W \times H}$, where the scalar and bivector components are kept to zero. The dimension 4 corresponds to the basis elements of $G(2,0,0)$, our chosen geometric algebra, consisting of $\{1, e_1, e_2, e_{12}\}$.

Inputs to and outputs from the second U-Net module are hence 2D vector fields, which can only be scaled and rotated by the layers working in $G(2,0,0)$. The initial number of flow proposals is downsampled to a single channel of optical flow, $\hat{\mathbf{v}}$, over which the loss is measured. The intuition behind GA-DDVM is similar to that of [45, 47] for 3D protein structures and camera poses, respectively: we want our network to make informed decisions about the object it is estimating by explicitly regressing those objects expressed as quantities in GA and by constraining the type of operations that can be performed on them.

### 4.1. Datasets

We evaluate GA-DDVM using AutoFlow for training, and KITTI and Sintel for testing. **AutoFlow** [62] is a large synthetic dataset (40k samples) designed for optical flow. It features diverse scenes with dense ground truth and is augmented following [11, 63]. **KITTI** [16] contains real-world driving scenes with ground truth optical flow from stereo LiDAR. It includes challenging outdoor conditions with fast motion, occlusions, and lighting variations. **Sintel** [7] is derived from the animated film "Sintel" and features complex, synthetic scenes with large displacements, non-rigid motion, and two variants: clean and final (with post-processing effects).

## 5. Experiments

We trained four models: (a) Open-DDVM (b) and Open-DDVM with the 4D correlation volume derived from [63] as well as (c) our own implementation of GA-DDVM without and (d) with correlation volume. The correlation vol-

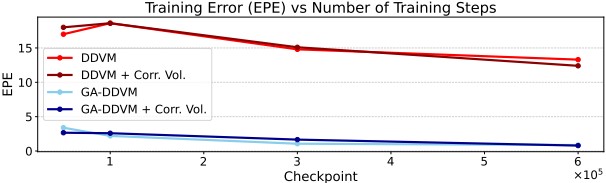

Figure 4. Training error (EPE) versus number of training steps for the four models tested.

ume is a tensor storing similarity scores between features from two images, which we employed as a prior similalry to [11]. The training parameters are kept identical to [11], with the exception of batch size, lowered from 16 to 8 for memory limitations, and learning rate, lowered from $10^{-4}$ to $10^{-5}$ to prevent the gradient from exploding. Just like Open-DDVM, the loss function we minimise is not that of Eq. 8, but the L2 norm between $\mathbf{v}, \hat{\mathbf{v}}$, which we found to work better in practice. We train the models across 4 A100 GPUs for an increasing number of steps namely 50 thousand, 100 thousand, 300 thousand and 600 thousand steps. The end-point error (EPE), defined as

$$\text{EPE} = \|\hat{\mathbf{v}}(\mathbf{x}) - \mathbf{v}(\mathbf{x})\|_2 \,, \tag{10}$$

is shown in Fig. 4 during training stage on the AutoFlow dataset.

Note how the addition of the correlation volume makes little to no difference in the accuracy of the pipeline. However, working in $G(2, 0, 0)$ is crucial: the EPE error with GA-DDVM reaches single digit pixel accuracy already just after 50 thousand iterations. DDVM, on the other hand, lags at EPE above 10 even after 600 thousand training steps.

The same pipelines have been tested over the KITTI and Sintel datasets, showing agreement with the trend shown during training. In Table 1 we report the EPE error on AutoFlow during training (labelled "AF") as shown in Fig. 4, as well as the EPE error on KITTI (labelled "K. EPE") and the Fl-all error (labelled "K. Fl-all"). The Fl-all error is shown in Eq. 11. It measures the proportion of pixels for which the estimated optical flow is significantly incorrect.

$$\text{Fl-all} = \frac{1}{N} \sum_{i=1}^{N} \mathbb{I}(\|\hat{\mathbf{v}}_i - \mathbf{v}_i\| > \Theta) \tag{11}$$

With $\Theta$ a threshold normally chosen to be 3 pixels or $5\%$, error above which a pixel is considered to be an outlier. The columns labelled as "S. cl" and "S. f" report the EPE for the Sintel clean and Sintel final datasets, respectively. The trend is clear: GA-DDVM, by operating on optical flows explicitly, is able to estimate the correct flow after few training iteration and generalise over previously unseen scenes.

Similarly to [11], we find that including the correlation volume has negligible impact on the performance of our GA-DDVM pipeline. This is likely because the geometric prior provides a stronger constraint than the correlation volume.

Examples of estimated optical flows with the four architectures are shown in Figs. 1-3 for the KITTI dataset and in Figs. 5-6 for the Sintel dataset. For all four example optical flows provided, is it clear to see that GA-DDVM is able to pick up early on the intensity of the optical flow, meaning it is able to identify which objects in the frame move more than others, as well as the colour, meaning it is able to detect the direction of motion early on. This is not true for Open-DDVM and Open-DDVM + correlation volume, for which the error is still significant after many iterations.

Table 2. Comparison of different models on Sintel and KITTI benchmarks. EPE = Endpoint error. Fl-all = Percentage of outlier pixels. * = as reported in [58], † = as reported in [11].

| Method | Pretraining | Iteration | Sintel Clean (EPE) | Sintel Final (EPE) | KITTI (Fl-all) |
|---|---|---|---|---|---|
| DDVM* | Palette-style | Unknown | 2.04 | 2.55 | 16.59% |
| Open-DDVM† (305k) | - | 305k | 2.96 | 3.97 | 20.38% |
| Open-DDVM† (900k) | - | 900k | 2.77 | 3.76 | 18.57% |
| Open-DDVM + Corr. Vol.† | - | 305k | 2.98 | 3.85 | 19.04% |
| GA-DDVM (ours) | - | 1M | 2.89 | 3.99 | 18.70% |
| GA-DDVM + Corr. Vol. (ours) | - | 1M | 2.87 | 3.88 | 18.44% |

Lastly, we trained GA-DDVM and GA-DDVM + correlation volume for 1M steps and compared them with the results reported in [58] and [11], as shown in Table 2. GA-DDVM slightly outperforms Open-DDVM on KITTI but performs similarly on Sintel. This is likely due to differences in training setups: Open-DDVM used different batch sizes and learning rates. Since diffusion models are sensitive to such hyperparameters, we expect GA-DDVM to show similar gains when trained under the same conditions, as suggested by Table 1.

Moreover, GA-DDVM is based on the Open-DDVM version of DDVM and, like Open-DDVM, does not outperform the original DDVM. This is partly because DDVM benefits from palette-style pretraining, which gives it an advantage over models like Open-DDVM and GA-DDVM that are trained from scratch.

## 6. Discussion

We argue that the faster convergence of GA-DDVM stems from the its second, specialized U-Net. We discuss the main points below.

**Task-Aligned Inductive Bias**. By restricting the network to rotations and scalings, GA-DDVM provides inductive bias that reflects the two main distortion in optical flow, i.e. directional and magnitude noise. At each timestep $t$, the model learns to denoise a noisy input $\mathbf{v}_t$ into the clean flow $\mathbf{v}_0$, which typically differs by an angle and scale. Rotor layers (Eq. 9) directly apply these corrections, focusing model capacity on the most relevant transformations.

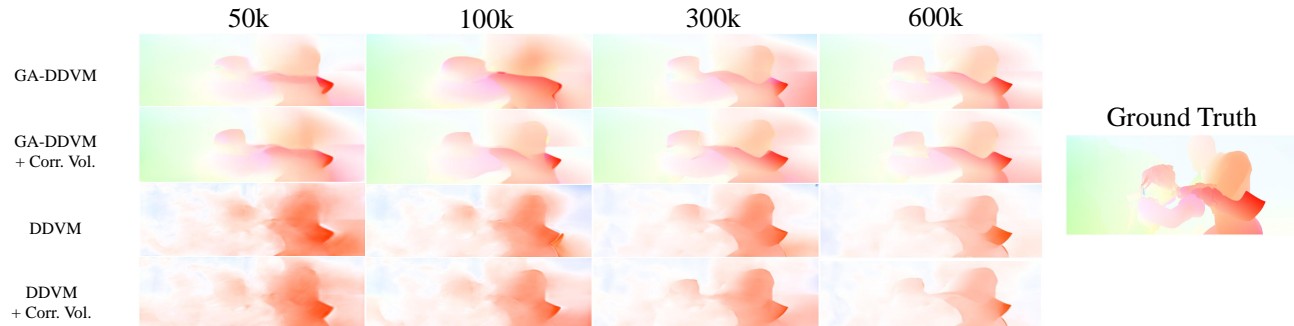

Figure 5. Ground truth and estimated optical flow for frame 160 in the Sintel dataset through four different architectures trained for an increasing number of steps.

**Efficient Hypothesis Space.** Convolutional layers constrain the hypothesis space by enforcing translation equivariance. This inductive bias reduces sample complexity and enhances generalization. Sandwich product layers, based on GA rotors, further restrict this space to transformations aligned with the underlying symmetries of the data, thereby enabling more physically meaningful representations and improved generalization capacity on the most relevant transformations [8].

**Avoiding Spurious Minima.** Rotor constraints eliminate non-geometric transformations (e.g., shear) that might fit training data but generalize poorly. This narrows the search space to physically plausible solutions, promoting monotonic convergence and better generalization.

**Fewer Parameters and Better Conditioning.** Each rotor is determined by a single angle. This parameter efficiency reduces overfitting and improves optimization. Moreover, rotors are unit-norm and represent orthogonal transformations, avoiding issues with poorly conditioned matrices and unstable gradients [55].

Whether adjusting direction or magnitude, the optimal transformation at any timestep into the diffusion pipeline is a rotation and scaling:

$$\hat{\mathbf{v}} = \alpha R(\theta)\hat{\mathbf{v}}_t, \tag{12}$$

where $\alpha = \|\mathbf{v}_0\|/\|\hat{\mathbf{v}}_t\|$ and $R(\theta)$ rotates $\hat{\mathbf{v}}_t$ into alignment with $\mathbf{v}_0$. Rotor layers can express this precisely, with the denoising function lying in:

$$\mathcal{F}_{\text{rot}} = \{f : \mathbf{v} \mapsto \alpha R\mathbf{v}\tilde{R}\}, \tag{13}$$

a subset of the general linear function space

$$\mathcal{F}_{\text{lin}} = \{f : \mathbf{v} \mapsto Wv\}, \tag{14}$$

enabling faster, targeted optimization.

**Decoupled Gradients.** The gradients of the loss with respect to the network parameters, specifically, the rotor angle $\phi$ and the scaling factor $\alpha$, influence orthogonal aspects of the predicted flow field: direction and magnitude, respectively. This property simplifies the optimization landscape. Below, we derive the gradients of the loss to illustrate this.

The loss we aim to minimize is the squared error between the predicted flow $\hat{\mathbf{v}}$ and the ground truth $\mathbf{v}_0$:

$$\mathcal{L} = \|\hat{\mathbf{v}} - \mathbf{v}_0\|^2. \tag{15}$$

The predicted flow is computed via a rotor-based transformation:

$$\hat{\mathbf{v}} = \alpha R_\phi \mathbf{u} \tilde{R}_\phi, \tag{16}$$

where $R_\phi = \cos(\phi/2) + \sin(\phi/2)e_1e_2$ is a rotor in $G(2,0,0)$ corresponding to a rotation by an angle $\phi$, and $\tilde{R}_\phi$ is its reverse. In 2D, this is equivalent to a rotation of the input vector $\mathbf{u} \in \mathbb{R}^2$ by angle $\phi$, giving:

$$\hat{\mathbf{v}} = \alpha \mathbf{R}_\phi \mathbf{u}, \tag{17}$$

where $\mathbf{R}_\phi \in \mathbb{R}^{2\times 2}$ is the rotation matrix:

$$\mathbf{R}_\phi = \begin{bmatrix} \cos\phi & -\sin\phi \\ \sin\phi & \cos\phi \end{bmatrix}. \tag{18}$$

Let $\mathbf{r}_\phi = \mathbf{R}_\phi \mathbf{u}$. The loss then becomes:

$$\mathcal{L} = \|\alpha\mathbf{r}_\phi - \mathbf{v}_0\|^2. \tag{19}$$

We first compute the gradient with respect to $\alpha$:

$$\frac{\partial \mathcal{L}}{\partial \alpha} = \frac{\partial}{\partial \alpha}\left(\alpha\mathbf{r}_\phi - \mathbf{v}_0\right)^T\left(\alpha\mathbf{r}_\phi - \mathbf{v}_0\right) \tag{20}$$

$$= 2(\alpha\mathbf{r}_\phi - \mathbf{v}_0)^T\mathbf{r}_\phi \tag{21}$$

$$= 2\alpha\|\mathbf{r}_\phi\|^2 - 2\mathbf{v}_0^T\mathbf{r}_\phi. \tag{22}$$

Let $\gamma$ denote the angle between the predicted flow $\hat{\mathbf{v}}$ and the ground truth $\mathbf{v}_0$. Then:

$$\mathbf{v}_0^T\mathbf{r}_\phi = \|\mathbf{v}_0\|\|\mathbf{r}_\phi\|\cos(\gamma) \tag{23}$$

so the gradient simplifies to:

$$\frac{\partial \mathcal{L}}{\partial \alpha} = 2\alpha\|\mathbf{r}_\phi\|^2 - 2\|\mathbf{v}_0\|\|\mathbf{r}_\phi\|\cos(\gamma) \tag{24}$$

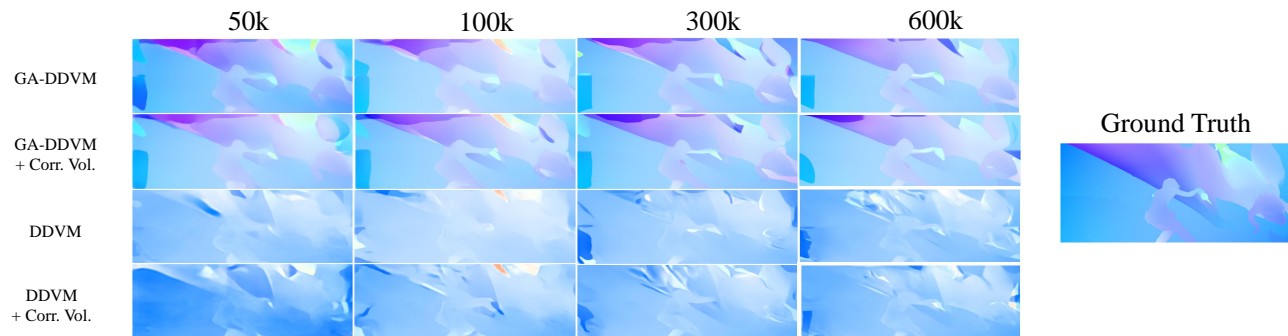

Figure 6. Ground truth and estimated optical flow for frame 500 in the Sintel dataset through four different architectures trained for an increasing number of steps.

Next, we compute the gradient with respect to the rotor angle $\phi$, which controls the direction of the predicted flow:

$$\frac{\partial \mathcal{L}}{\partial \phi} = 2\alpha(\alpha\mathbf{r}_\phi - \mathbf{v}_0)^T \frac{d\mathbf{R}_\phi}{d\phi}\mathbf{u}, \qquad (25)$$

this expression captures how rotating the input by $\phi$ changes alignment with the target vector $\mathbf{v}_0$. In summary:

$$\frac{\partial \mathcal{L}}{\partial \alpha} \propto \alpha - \frac{\|\mathbf{v}_0\|}{\|\mathbf{r}_\phi\|}\cos(\gamma) \qquad (26)$$

$$\frac{\partial \mathcal{L}}{\partial \phi} \propto \|\mathbf{v}_0\|\|\mathbf{u}\|\sin(\gamma) \qquad (27)$$

where $\gamma$ is the angular error between $\hat{\mathbf{v}}$ and $\mathbf{v}_0$. Eq. 26 depends only on the difference in magnitude, scaled by the cosine of the angle $\gamma$ between prediction and ground truth. Similarly, Eq. 27 depends only on the angular misalignment between prediction and ground truth, it does not involve $\alpha$ directly.

## 7. Conclusions

We introduced GA-DDVM, one of the earliest reported diffusion pipelines to operate within Geometric Algebra (GA), aimed at improving optical flow estimation. By embedding data and transformations directly in GA, GA-DDVM constrains both the representation (2D vector fields) and the transformations applied (scaling and rotation), allowing the network to focus its capacity on physically meaningful variations. Built on a W-Net backbone with GCAN layers, GA-DDVM demonstrates a substantial reduction in training overhead: it achieves strong optical flow predictions, capturing both magnitude and direction, within just 50,000 training steps and without any pretraining. This suggests that GA-DDVM can significantly reduce the sample and compute complexity of training diffusion models for geometric tasks. Preliminary comparisons show it outperforms the open DDVM implementation under identical conditions. We anticipate that aligning the

training schedule (e.g., increasing batch size and learning rate) to match that of Open-DDVM will further enhance performance. GA-DDVM offers a promising path toward more efficient and cost-effective training of diffusion-based vision models by embedding inductive biases directly into the network structure.

**Limitations.** An important limitation of GA-DDVM is its slower training time, up to $2\times$ slower compared to DDVM. This is due to the current tensor-based implementation of GA, as seen in [5, 26, 41]. This overhead is purely a function of the implementation and not intrinsic to the method; it is a known fact the literature and is considered the trade-off for the improved convergence speed and inductive bias offered by geometric structure. Sampling time remains identical to that of non-GA diffusion models. A more extensive training strategy is due: the lack of pretraining or hyperparameter sweep limits the strength of comparative evaluations to previously reported results.

**Future work.** This work serves as a proof of concept, introducing one of the earliest diffusion pipelines in GA for structured prediction. Future research could extend this framework to a wide range of tasks, including monocular depth estimation, surface normals, 3D motion, and broader applications such as PDE solvers or protein modeling, fields where data can be naturally embedded as geometric primitives or multivectors. Another possible direction is the joint estimation of multiple correlated quantities within a unified GA-based architecture, enabling more robust and efficient multi-task learning (e.g., depth and optical flow as scalar and vector components of $G(2, 0, 0)$, as done in [17]).

GA networks have the potential to play a key role in geometrically principled diffusion models, offering significantly faster convergence by aligning learning with the symmetries and constraints of the problem in a strikingly simple yet mathematically rigorous way.

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
