# OpenReview forum: "Geometric Inductive Priors in Diffusion-Based Optical Flow Estimation"
_thecvf.com/ICCV/2025/Workshop/BEW — BEW 2025 Oral_

### Official Review · Reviewer_8zZF · 2025-07-05
**Review for "Geometric Inductive Priors in Diffusion-Based Optical Flow Estimation"**

**Rating:** 2
**Confidence:** 4

**Review:**

**Summary:**

The authors propose GA-DDVM, a trainable sub-network for performing geometric operations on optical flow constrained to rotations and scaling. Experiments on optical flow estimation show faster training convergence as compared to a pure diffusion model-based pipeline.

**Strengths:**

The paper goes in depth with the theoretical foundations for using Geometric algebra, making it easy to read. The novelty of using the proposals of optical flow from the penultimate layer of DDVM and refining them using rotations and scaling operations using steerable layers is clearly defined. This can, as an extension, also be used to refine optical flow predictions from multiple candidates. The performance of the combined pipeline shows nice results and paves the way for faster converging diffusion model pipelines in the future.

**Weakness:**
- While the idea of generating candidate proposals from diffusion models due to its probabilistic nature and refining them using Clifford Steerable layers is unique, using Clifford algebra on optical flow and depth is not new [1] (*also missing in the related work section*), where authors use both optical flow and depth signal to jointly model the respiratory waveform.

- Continuing on this, while the original work [2] also estimates depth, this paper lacks experiments on depth estimation, limiting the application of the model to estimating optical flow. Directions similar to [1] can be used to jointly model both quantities.

- In Table 2, GA-DDVM at 1M steps performs similarly to Open-DDVM for zero-shot optical flow estimation on KITTI/Sintel, raising questions about its claimed generalization advantage on unseen scenes.

- No explanation is provided as to why using GA-DDVM with correlation volume makes little to no difference in EPE error.

- No analysis on the memory overhead as compared to the baseline.

- Reported metrics differ significantly across tables. For example, at ~300k steps, DDVM's Sintel Clean EPE is 10.9 in Table 1 but 2.96 for Open-DDVM in Table 2 as reported in the original work. This may be due to different hyperparameters for both studies, but suggests the sub-optimal setup for comparing the performance of GA-DDVM vs. vanilla DDVM in Table 1.

- At 100k training steps, the plot (Figure 4) shows the End-Point Error (EPE) around 15 for both DDVM and DDVM with correlation volume, whereas the corresponding values in Table 1 for the AF column are around 18.5. This mismatch needs clarification.

- The values of EPE of GA-DDVM and GA-DDVM + Corr. Vol. seems to juggle in Table 1, whereas in the plot, GA-DDVM + Corr. Vol. always seems to have less error.

- The authors trained four models, Open-DDVM and GA-DDVM, along with their correlation volume counterparts. Models are trained for a progressive number of steps (50k, 100k, 300k, 600k). However, the plot legend in the figures and tables uses *DDVM* instead of *Open-DDVM*, which is inconsistent with the text and confusing.

- It is mentioned in the text that the authors use AutoFlow for training and KITTI/Sintel for testing. For Table 1, it's unclear if models were trained from scratch on KITTI/Sintel or only first trained on AutoFlow and evaluated zero-shot on KITTI/Sintel. This ambiguity affects the interpretation of Table 1, as it shows performance with respect to training steps.






[1] Ghezzi, Omar, et al. "CliffPhys: Camera-Based Respiratory Measurement Using Clifford Neural Networks." European Conference on Computer Vision. Cham: Springer Nature Switzerland, 2024.

[2] Saxena, Saurabh, et al. "The surprising effectiveness of diffusion models for optical flow and monocular depth estimation." Advances in Neural Information Processing Systems 36 (2023): 39443-39469.

[3] Dong, Qiaole, Bo Zhao, and Yanwei Fu. "Open-ddvm: A reproduction and extension of diffusion model for optical flow estimation." arXiv preprint arXiv:2312.01746 (2023).

---

### Official Review · Reviewer_qiK3 · 2025-07-06
**A valuable work on estimating optical flow with a Clifford Algebra-based network**

**Rating:** 4
**Confidence:** 3

**Review:**

This paper tackles optical flow estimation by exploiting the framework of Geometric Algebra incorporated in a Denoising Diffusion Probabilistic Model pipeline. The method is benchmarked on KITTI and Sintel. The presented results give some interesting insights into non-Euclidean geometrical constraints.

The paper is fairly written, and the method is rigorously presented and discussed. Technical background on Clifford Algebra is given through the discussion, and the reader is guided toward effective references to deepen their understanding. The empirical results, while not markedly superior to those of the baselines, are promising. While theoretically valid, the method's efficiency is tight to the current implementations, which makes it remarkably slower than state-of-the-art techniques, limiting its applicability, as fairly pointed out by the authors. In light of this limitation, the authors' claim about faster convergence is cut back.

Some minor flaws should be addressed. The end of Section 2 (from L.147 on) would be better placed as a subsection of Section 3, as it presents some preliminary knowledge about GA, more than a literature review. The statement at L. 395-402 lacks some reference. The readability and clarity of Table 2 could be improved by marking the results that should be read and compared together, and highlighting the best results.

Overall, this paper provides an interesting analysis of GA-based networks. While limited by computational efficiency, I think this work could be valuable for the workshop.

---

### Official Review · Reviewer_cPFr · 2025-07-07
**The authors incorporate geometric algebra priors into DDVM networks, opening up new paths for directly using geometric inductive biases in the models. Therefore, I believe this would be a valuable addition to the research community working on incorporating various geometries into deep learning networks.**

**Rating:** 4
**Confidence:** 4

**Review:**

Summary: The authors propose that constraining DDVMs with geometric algebra priors can enhance optical flow estimation by making the network faster and more accurate. They achieve this by restricting the type of object learned by the network to 2D vector fields for optical flows, which leads to faster convergence. Additionally, they limit the operations performed by the network layers on these objects to scaling and rotations, thereby reducing the number of network parameters. They also claim that this approach boosts interpretability by working explicitly in 2D space.

Strengths: By embedding both the data and the network architecture within a Geometric Algebra (GA) framework, the authors' approach opens up promising avenues for utilizing GA in architectures such as UNet and DDVMs. The network architecture design is well explained in the paper, making it intuitive for readers. The benefits of incorporating GA are discussed and are theoretically supported by derivations.

Weaknesses: Despite the strengths mentioned above, the authors restrict themselves to G(2,0,0) due to the chosen task of optical flow estimation, which still lies within Euclidean space. While the vision of the paper aligns well with the goals of the workshop, the continued focus on Euclidean space is the main reason I assign a rating of 4 to the paper.

Questions/clarifications:
1. Two versions of the GA-DDVM model are discussed -- one with a correlation volume and one without. However, it is unclear what the GA-DDVM with a correlation volume entails. Please update the text to include the relevant implementation details.
2. The authors claim that their approach enhances interpretability by grounding both the data and the network architecture in geometric priors. However, no experiments are presented to support this claim. Could the authors include experiments involving only rotational or scaling motions in videos, demonstrating that the network can estimate optical flow in such scenarios with high accuracy?

---

### Decision · Program_Chairs · 2025-07-09

**Decision:**

Accept (Oral)

**Comment:**

The majority of the reviews agree towards accepting the paper.  The authors should do their best to address the comments of the reviewers in their final version. The oral presentations would also present a poster.